# *Mycobacterium leprae* promotes triacylglycerol *de novo* synthesis through induction of GPAT3 expression in human premonocytic THP-1 cells

Kazunari Tanigawa[1], Yasuhiro Hayashi[2], Kotaro Hama[3], Atsushi Yamashita[2], Kazuaki Yokoyama[3], Yuqian Luo[4], Akira Kawashima[4], Yumi Maeda[5], Yasuhiro Nakamura[1], Ayako Harada[1], Mitsuo Kiriya[4], Ken Karasawa[1], Koichi Suzuki[4,5]*

**1** Department of Molecular Pharmaceutics, Faculty of Pharma-Science, Teikyo University, Itabashi-ku, Tokyo, Japan, **2** Department of Biological Chemistry, Faculty of Pharma-Science, Teikyo University, Itabashi-ku, Tokyo, Japan, **3** Department of Physical Pharmaceutics, Faculty of Pharma-Science, Teikyo University, Itabashi-ku, Tokyo, Japan, **4** Department of Clinical Laboratory Science, Faculty of Medical Technology, Teikyo University, Itabashi-ku, Tokyo, Japan, **5** Department of Mycobacteriology, Leprosy Research Center, National Institute of Infectious Diseases, Higashimurayama-shi, Tokyo, Japan

\* koichis0923@med.teikyo-u.ac.jp

**Data Availability Statement:** All relevant data are within the manuscript and its Supporting Information files.

## Abstract

*Mycobacterium leprae* (*M. leprae*) is the etiological agent of leprosy, and the skin lesions of lepromatous leprosy are filled with numerous foamy or xanthomatous histiocytes that are parasitized by *M. leprae*. Lipids are an important nutrient for the intracellular survival of *M. leprae*. In this study, we attempted to determine the intracellular lipid composition and underlying mechanisms for changes in host cell lipid metabolism induced by *M. leprae* infection. Using high-performance thin-layer chromatography (HPTLC), we demonstrated specific induction of triacylglycerol (TAG) production in human macrophage THP-1 cells following *M. leprae* infection. We then used [14C] stearic acid tracing to show incorporation of this newly synthesized host cell TAG into *M. leprae*. In parallel with TAG accumulation, expression of host glycerol-3-phosphate acyltransferase 3 (GPAT3), a key enzyme in *de novo* TAG synthesis, was significantly increased in *M. leprae*-infected cells. CRISPR/Cas9 genome editing of GPAT3 in THP-1 cells (*GPAT3* KO) dramatically reduced accumulation of TAG following *M. leprae* infection, intracellular mycobacterial load, and bacteria viability. These results together suggest that *M. leprae* induces host GPAT3 expression to facilitate TAG accumulation within macrophages to maintain a suitable environment that is crucial for intracellular survival of these bacilli.

## Introduction

Leprosy is a chronic infectious disease caused by *Mycobacterium leprae* (*M. leprae*), which mainly affects the skin and peripheral nerves. In 2018, 208,641 new cases of leprosy were

**Funding:** This work was supported by AMED under Grant Numbers JP17fk0108303 (to K.S.) and JP20fk0108064 (to K.S.), MEXT KAKENHI Grant Numbers 15K190097 (to K.T.) and 18K15150 (to K.T.), and Sasakawa Scientific Research Grant Number 26-428 (to K.T.).

**Competing interests:** The authors have declared that no competing interests exist.

reported in 127 countries worldwide [1]. Leprosy is a neglected tropical disease (NTD) that is classified into two forms based on clinical, histological and bacteriological features [2]: lepromatous leprosy and tuberculoid leprosy. Lepromatous leprosy is a progressive and disseminated disease characterized by widespread skin lesions in which bacilli undergo unrestricted multiplication inside foamy histiocytes. In these lesions, *M. leprae* replicates within enlarged, lipid-filled phagosomes that facilitate its infectious activity [3].

The surface of lipid droplets is decorated by proteins, particularly adipose differentiation-related protein (ADRP) and perilipin, which are involved in regulation of lipid metabolism [4, 5]. We previously reported that ADRP and perilipin are highly expressed in foamy macrophages in dermal leprosy granulomas [6]. Consistent with *in vivo* data, ADRP and perilipin are also strongly induced by *M. leprae* infection and localize within *M. leprae*-containing phagosomes in human macrophages *in vitro* [6]. Meanwhile, *M. leprae* suppresses degradation of lipid droplets by suppressing expression of hormone-sensitive lipase (HSL), thus contributing to a lipid-rich environment in host macrophages [7]. These results suggest that *M. leprae* infection profoundly modifies host cell lipid metabolism. However, the exact composition of lipid droplets in *M. leprae*-infected cells is not well defined and which lipids are actually utilized by *M. leprae* is not known.

*M. leprae* is reported to increase expression of cholesterol synthase (HMG-CoA reductase) in host cells, whereas inhibition of *de novo* cholesterol synthesis by lovastatin reduces mycobacterial viability [8]. *Mycobacterium tuberculosis* (*M. tuberculosis*), which uses host cell-derived cholesterol as a carbon source, incorporates cellular lipids into bacterial cell membranes via the Mce4 transporter [9]. However, *M. leprae* has lost the *mce4* operon and thus cannot use cholesterol directly as a nutrient source. In order to survive, *M. leprae* may instead actively convert cholesterol to cholestenone via cholesterol oxidase (ML1492) [10]. In Schwann cells, *M. leprae* stimulates glucose uptake and activates the pentose phosphate pathway that contributes to triacylglycerol (TAG) synthesis [11].

Together, these findings demonstrate the importance of host-derived lipids for mycobacteria parasitism. Thus, a comprehensive analysis of *M. leprae*-induced lipid droplets in macrophages and elucidation of how *M. leprae* utilizes host-derived lipids can contribute to a better understanding of its parasitism mechanism. In this study, we examined changes in intracellular lipid composition induced by *M. leprae* infection, as well as underlying molecular mechanisms in macrophages that are involved in these changes.

## Materials and methods

### *M. leprae* preparation and cell culture

The *M. leprae* were grown in the footpads of nude mice and prepared at the Leprosy Research Center, National Institute of Infectious Diseases, Tokyo, Japan as described previously [12–14]. The human premonocytic cell line THP-1 was obtained from the American Type Culture Collection (ATCC; Manassas, VA). Cells were cultured in 10-cm tissue culture dishes in RPMI-1640 medium supplemented with 10% charcoal-treated fetal bovine serum (FBS) and 50 mg/ml penicillin/streptomycin at 37°C and 5% $CO_2$. THP-1 cells ($3 \times 10^6$) were treated with latex beads (Fluoresbrite microspheres; Technochemical, Tokyo, Japan), peptidoglycan (Sigma, St Louis, MO) or live or heat-killed (80°C, 30 min) bacilli ($1.5 \times 10^8$) at a multiplicity of infection (MOI) of 50. The animal experiment was reviewed and approved by the Experimental Animal Committee of the National Institute of Infectious Diseases (Permit No. 118028), and all experiments were conducted according to the recommended guidelines.

## Lipid analysis by High-Performance Thin-Layer Chromatography (HPTLC)

THP-1 cells ($3 \times 10^6$) were treated with either latex beads (Fluoresbrite microspheres; Techno-chemical, Tokyo, Japan), peptidoglycan (Sigma, St Louis, MO), or $1.5 \times 10^8$ bacilli, which were either live or heat-killed, at a multiplicity of infection (MOI) of 50. After treatment, lipids were extracted from the cells using the Bligh and Dyer method [15], and the solutions were evaporated to dryness. The dried lipids were dissolved in 10 µl chloroform/methanol (2:1, v/v), and applied to silica gel 60 HPTLC plates ($10 \times 10$ cm, Merck, Darmstadt, Germany) that were developed in hexane/ethyl ether/acetic acid (60:40:1, v/v/v) until the solvent front reached the middle of the plate. The plates were then further developed in hexane/chloroform/acetic acid (80:20:1, v/v/v). After separation, the plates were stained by spraying a charring solution containing 10% $CuSO_4$ and 8% $H_3PO_4$ and heated at 180˚C for 10 min. Lipid bands were quantified by photo densitometry using ImageJ/Fiji software (https://imagej.net/Fiji).

## RNA preparation and quantitative real-time PCR (qRT-PCR)

RNA from THP-1 cells was prepared using an RNeasy Plus Mini Kit (Qiagen Inc., Valencia, CA) as described previously [6, 13]. The concentrations and quality of total RNA samples were evaluated with an e-Spect spectrophotometer (BMBio, Tokyo, Japan). Total RNA from each sample was reverse-transcribed to cDNA using a High Capacity cDNA Reverse Transcription Kit (Applied Biosystems, Foster City, CA) as described previously [6, 13]. Primers used for PCR to amplify specific cDNAs are listed in S1 Table. Quantitative real-time PCR (qRT-PCR) was performed on an ABI 7500 Fast system using SYBR Select Master Mix (Applied Biosystems). Target genes were normalized to the *β-actin* level in each sample. Quantitative measurements were performed using the ΔΔCt method.

## Protein preparation and Western blot analysis

Cellular protein was extracted and analyzed as previously described [16, 17]. Briefly, cells were washed three times with ice-cold PBS and lysed in lysis buffer containing 50 mM HEPES, 150 mM NaCl, 5 mM EDTA, 0.1% NP40, 20% glycerol, and protease inhibitor cocktail (Complete Mini, Roche, Indianapolis, IN) for 1 h. Cells were then transferred to a 1.5 ml tube and centrifuged. The supernatant was transferred into a new tube, and then the protein concentration was measured. Proteins (10 µg) were heated in SDS sample loading buffer at 70˚C for 10 min and loaded onto NuPage 4–12% Bis-Tris gels (Thermo Fisher Scientific). After electrophoresis, proteins were transferred to a PVDF membrane using an iBlot 2 transfer system (Thermo Fisher Scientific). The membrane was washed with PBST (PBS with 0.1% Tween 20), blocked overnight with ImmunoBlock (KAC Co., Ltd., Kyoto, Japan), and then incubated with anti-GPAT3 (Thermo Fisher Scientific; 1:2,000 dilution) or anti-β-actin (Cell Signaling Technology, Danvers, MA; 1:2,000 dilution) antibodies. After washing with PBST, the membrane was incubated for 1 h with HRP- conjugated goat anti-rabbit IgG secondary antibody (Thermo Fisher Scientific; 1:1,500 dilution). The signal was developed using Amersham ECL Prime (GE Healthcare, Buckinghamshire, UK), and the images were scanned with an Amersham Imager 680 RGB (GE Healthcare).

## Generation of CRISPR/Cas9-based GPAT3 Knockout (KO) cells

To establish *GPAT3* KO THP-1 cells, guide RNAs were designed using target design software developed by Dr. Feng Zhang's group at the Massachusetts Institute of Technology (http://crispr.mit.edu). The guide RNA sequence, 5′-ATGGAGGGCGCAGAGCTGGC-3′, was cloned

into the pSpCas9(BB)-2A-GFP (PX458) vector (Addgene, Watertown, MA). The construct was then transfected into THP-1 cells using the Xfect Transfection Reagent (Clontech Laboratories, Inc., Mountain View, CA) according to the manufacturer's instructions. The transfected cells were cultured for 48 h and GFP-positive cells were selected using a FACSAria III cell sorter (BD Biosciences, Franklin Lakes, NJ). Clonal populations of *GPAT3* KO cells were isolated using limiting dilution. Disruption of the *GPAT3* gene was confirmed by DNA sequencing and protein expression.

## Immunofluorescence staining

THP-1 cells were infected with FITC-conjugated *M. leprae* and cultured on glass-bottom dishes (Matsunami Glass, Osaka, Japan) for 24 h. After discarding the supernatant, the cells were washed with PBS to remove excess extracellular *M. leprae*, fixed with 3% paraformaldehyde (Wako Pure Chemicals, Osaka, Japan) in PBS for 15 min, then permeabilized with 0.1% Triton X-100 (Wako Pure Chemical) in PBS for 10 min. After treating with blocking buffer (ImmunoBlock) for 10 min, the cells were incubated with HCS LipidTOX Red neutral lipid stain (Thermo Fisher Scientific; 1:1,000 dilution) with blocking buffer for 30 min at room temperature in the dark. After washing with PBS, the nuclei were counterstained with Hoechst 33258 (Thermo Fisher Scientific; 1:2,000 dilution) with blocking buffer for 10 min at room temperature. Immunofluorescence was visualized and the images were captured with a FV10i confocal laser-scanning microscope (Olympus, Tokyo, Japan). Quantification of LipidTOX staining was performed by tracing the region-of-interest (ROI) to generate fluorescence intensity values using FV10i software (Olympus).

## Flow cytometry

THP-1 cells ($1 \times 10^5$) cultured in 6-well plates were infected with FITC-conjugated *M. leprae* (MOI: 0, 50, 100, 200 and 500), and incubated for 24 h. The culture medium was discarded, the cells were washed three times with 3 ml of warm PBS to remove extracellular *M. leprae*, and then fixed with 3% buffered formalin. Cells were suspended in PBS with 1 mM EDTA to analyze the fluorescence intensity using a FACSCanto II flow cytometer (BD Biosciences) and FlowJo software (FlowJo LLC, Ashland, OR).

## Metabolic labeling of TAG

THP-1 cells ($1 \times 10^6$) were inoculated with ether live or heat-killed *M. leprae* (MOI: 10 and 50) for 24 h, then incubated with 0.2 μCi [$^{14}$C] stearic acid (American Radiolabeled Chemicals, Saint Louis, MO) for 16 h at 37°C with 5% $CO_2$. The cells were washed three times with 0.25% BSA in PBS and suspended in 0.05% Tween80/HBSS. After homogenization by 20 passages through a 22G needle, the homogenates were centrifuged at 250×*g* for 10 min at 4°C. The supernatant was collected in a new tube, 1/5 volume of 0.25% trypsin was added, and the mixture was incubated for 1 h at 37°C. After centrifugation, the supernatant was discarded. The pellet was vortexed with 0.05% Tween80/HBSS prior to the addition of 0.5N NaOH and incubation at 37°C for 15 min. After incubation, *M. leprae* was washed twice with 0.05% Tween/HBSS by centrifugation at 3,000×*g* for 20 min at 4°C. The presence of *M. leprae* was confirmed by PCR using a primer for *hsp70*. *M. leprae*-derived lipids were prepared as described above and applied to a TLC plate, which was developed with hexane/ether/acetic acid (80/30/1, v/v/v). Radioactive TAG on the TLC plate was detected with a Typhoon FLA 9500 instrument (GE Healthcare) and quantified using ImageQuant TL software (GE Healthcare).

### Quantification of *M. leprae* RNA expression

THP-1 cells ($3 \times 10^6$) were infected with live *M. leprae* (MOI: 50) for 24 h and *M. leprae* was purified as described above. RNA was extracted using an RNeasy Plus Mini Kit (Qiagen) and reverse-transcribed to cDNA using a High Capacity cDNA Reverse Transcription Kit (Applied Biosystems) as described previously [18]. PCR primers to amplify cDNAs are listed in S1 Table. Touchdown PCR was performed using a Thermal Cycler Dice (Takara) as described previously [7, 19]. The PCR products were analyzed by 2% agarose gel electrophoresis.

### Statistical analysis

Statistical analyses were performed with GraphPad PRISM 6 (GraphPad Software). For comparisons, Student's *t* test or one-way analysis of variance (ANOVA) followed by the Dunnett test were used. The results are shown as the mean ± S.D.

## Results

### *M. leprae* promoted TAG accumulation in THP-1 cells

To clarify the effect of *M. leprae* infection on the host cell lipid composition, we first examined the lipid components of *M. leprae*-infected THP-1 cells using HPTLC conditions suitable for separating neutral lipids [8]. Heat-killed *M. leprae*, latex beads and peptidoglycan were used as controls. The positions of cholesterol ester (ChoE), triacylglycerol (TAG), fatty acid (FA), cholesterol (Cho), diacylglycerol (DAG), monoacylglycerol (MAG) and phospholipid (PL) were determined by their relative to front (RF) values based on the position of standard lipids (S1 Fig).

The results showed that the absolute amount of TAG visualized on the TLC plates increased only after a live *M. leprae* infection (Fig 1A). TAG accumulation was transiently induced by

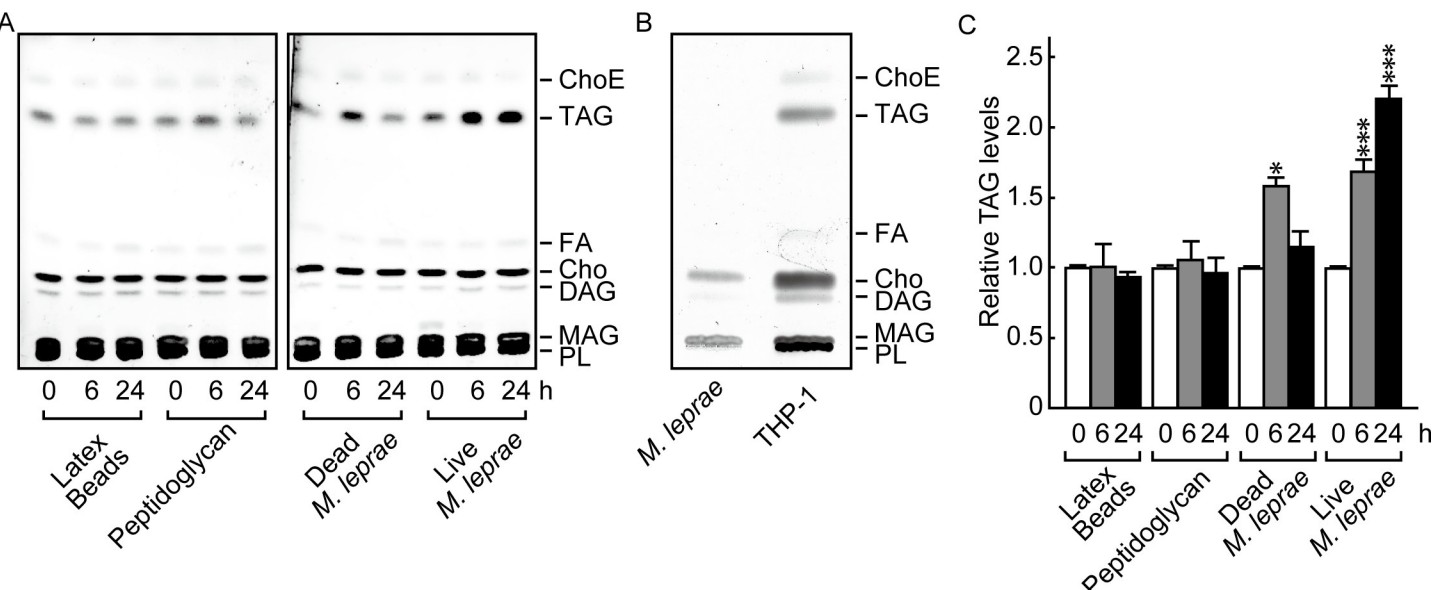

**Fig 1. *M. leprae* infection enhanced TAG accumulation in THP-1 cells.** (A) THP-1 cells ($3 \times 10^6$) were cultured in 6-well plates with live *M. leprae* (MOI: 20), heat killed (80°C for 30 min) *M. leprae*, 2 μg/ml peptidoglycan, or latex beads for the indicated period of time. Total lipids were extracted and analyzed by HPTLC. (B) Total lipids extracted from *M. leprae* and THP-1 cells were analyzed by TLC. (C) The amount of TAG as measured by densitometry and expressed relative to levels at 0 h. Values represent the mean ± S.D. from three independent experiments. Significance was determined by a one-way ANOVA followed by a Dunnett test. One and three asterisks indicate *p*<0.05 and *p*<0.001, respectively. ChoE: Cholesterol ester; TAG: Triacylglycerol; FA: Fatty acid; Cho: Cholesterol; DAG: Diacylglycerol; MAG: Monoacylglycerol; PL: Phospholipid.

dead bacilli at 6 h, but the levels returned to baseline within 24 h (Fig 1A). The observed transient TAG induction corresponds with previous observations that heat-killed *M. leprae* changed host mRNA levels of ADRP, perilipin, and HSL [6, 7]. Cells treated with latex beads or peptidoglycan showed no significant change in the amount of TAG (Fig 1A). Meanwhile, the absolute amount of TAG visualized on TLC plates was clearly increased after live *M. leprae* infection (Fig 1A). Since TAG was not detected in the lipids derived from *M. leprae* before infection (Fig 1B), it suggests that the observed lipids were derived from host cells.

We also measured the density of each lipid and calculated the relative proportion of TAG among all the lipid types. The proportion of TAG among the overall lipid population was also significantly increased by *M. leprae* infection (Fig 1C). Other lipid components, such as PL, FA, Cho and ChoE, showed no change even after live *M. leprae* infection (data not shown). These results suggest that TAG is the key lipid that is specifically induced by *M. leprae* infection in host macrophages.

## Glycerol-3-phosphate acyltransferase 3 (GPAT3) expression was induced in THP-1 cells following *M. leprae* infection

Glycerol-3-phosphate acyltransferase (GPAT) is a rate-limiting enzyme in TAG biosynthesis and is present in four isoforms: GPAT1, GPAT2, GPAT3 and GPAT4 [20–23]. To investigate the potential involvement of these GPAT isoforms in *M. leprae*-infected host cells, we used qRT-PCR to examine the mRNA expression levels of *GPAT1*, *GPAT2*, *GPAT3* and *GPAT4* in THP-1 cells after *M. leprae* infection. Endogenous GPAT isoforms were expressed at a similar level in THP-1 cells (S2 Fig). Among the four isoforms, only *GPAT3* mRNA levels increased after *M. leprae* infection at MOI 10 and 20 in 24 h (Fig 2A). The increase in *GPAT3* mRNA was apparent as early as 6 h after *M. leprae* infection and increased up to 48 h post-infection (Fig 2B). In accordance with mRNA levels, Western blotting showed that the amount of GPAT3 protein expression also increased following *M. leprae* infection (Fig 2C).

To clarify whether the observed increase in GPAT3 expression was specific for *M. leprae* infection, or instead was due to non-specific effects accompanying phagocytosis of particles and/or macrophage activation, we examined the effects of dead *M. leprae*, latex beads and peptidoglycan on *GPAT3* expression. Neither latex beads nor peptidoglycan affected *GPAT3* mRNA expression, but treatment of cells with heat-killed *M. leprae* transiently increased *GPAT3* mRNA levels at 6 h before they returned to their original levels by 24 h (Fig 2D). The sustained induction of *GPAT3* by live *M. leprae* and transient induction of *GPAT3* by dead *M. leprae* were in accordance with changes in intracellular TAG accumulation shown in Fig 1, suggesting that GPAT3 is the key molecule that mediates increases in the amount of TAG following *M. leprae* parasitism.

## *M. leprae* induces formation of lipid droplets through GPAT3

To further evaluate the effect of GPAT3 on *M. leprae*-induced accumulation of TAG, we generated *GPAT3* KO THP-1 cells using the CRISPR/Cas9 gene-editing system. Among the three KO clones isolated, we used one clone with a 16-nucleotide deletion in exon 1 for subsequent experiments. This deletion produced a homozygous deletion of *GPAT3* with a frameshift in both alleles (Fig 3A). The lack of GPAT3 protein in the KO clone was confirmed by Western blot (Fig 3B). We then compared the effect of *M. leprae* infection on lipid droplet formation in wild-type and *GPAT3* KO cells using LipidTOX staining (Fig 3C) and quantified the LipidTOX fluorescence intensity in cells (Fig 3D). We observed a large number of lipid droplets in wild-type cells following *M. leprae* infection, whereas the number of lipid droplets was reduced by about 80% in *GPAT3* KO cells, compared to the wild-type cells. Although both GPAT3 and

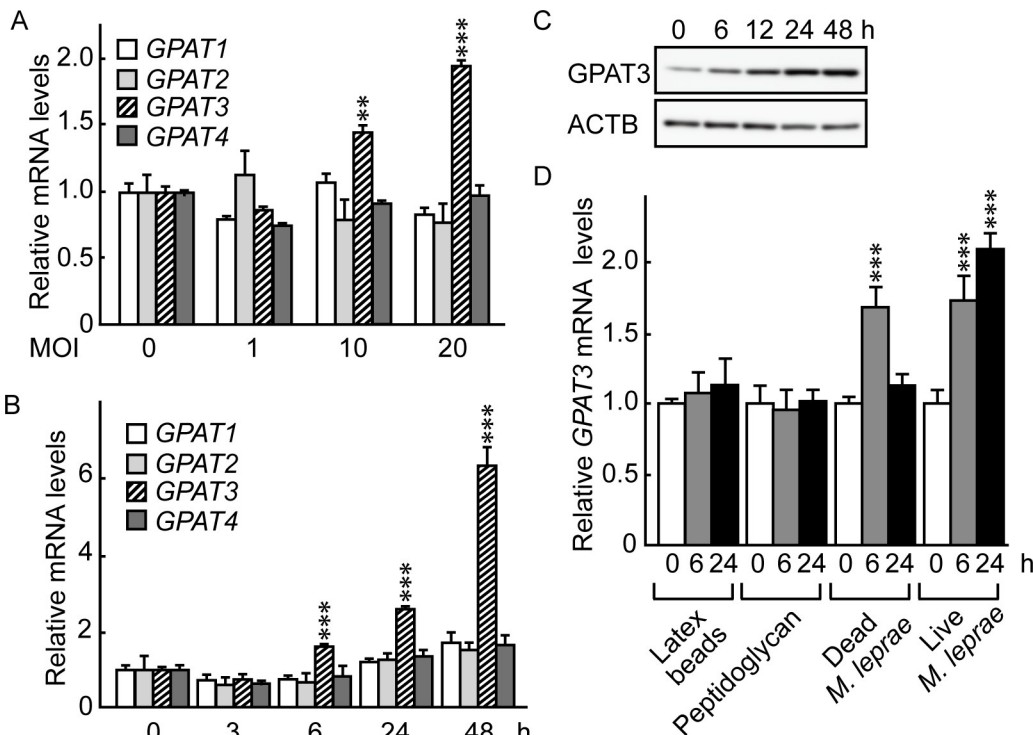

**Fig 2. *M. leprae* infection induced GPAT3 expression.** (A) THP-1 cells ($3 \times 10^6$) were cultured in 6-well plates and inoculated with *M. leprae* (MOI: 0, 1, 10 and 20). After a 24 h incubation, total RNA was purified and the expression level of GPAT isoforms was evaluated by qRT-PCR. Expression levels of mRNA were normalized relative to *β-actin* (*ACTB*) levels and are expressed relative to their levels at MOI 0. (B and C) THP-1 cells were cultured in 6-well plates and inoculated with *M. leprae* (MOI: 20). Total RNA and protein were purified from the cells for (B) qRT-PCR and (C) Western blotting analysis. *GPAT* mRNA levels were normalized against *ACTB* and are expressed relative to its level at 0 h. (D) THP-1 cells were cultured with live *M. leprae*, dead *M. leprae*, 2 μg/ml of peptidoglycan, or latex beads for the indicated period of time. Total RNA was purified and subjected to an qRT-PCR analysis. Statistical significance was determined by a one-way ANOVA followed by a Dunnett test. Two and three asterisks indicate $p<0.01$ and $p<0.001$, respectively. The representative results of three independent experiments are shown.

GPAT4 are microsomal enzymes that share similar functions, the expression of *GPAT4* in *GPAT3* KO cells did not change, even after *M. leprae* infection (S3 Fig), confirming that GPAT3 is the main isoform that responds to *M. leprae* infection (Fig 2).

To determine whether *M. leprae* utilizes the newly synthesized TAG that exists in lipid droplets, we performed metabolic labelling of THP-1 cells with [$^{14}$C] stearic acid to follow *de novo* synthesis of TAG. Twenty-four hours after *M. leprae* inoculation of wild-type and *GPAT3* KO THP-1 cells, [$^{14}$C] stearic acid was added, and the cells were cultured for 16 h. Then, radioactivity of TAG in the bacilli was examined on a TLC plate. The presence of bacilli isolated from *M. leprae*-infected cells was confirmed by expression of *M. leprae*-specific *hsp70* gene expression (Fig 3E). Radioactive TAG was detected only in *M. leprae* extracted from wild-type cells, but not from *GPAT3* KO cells (Fig 3F). While strong radioactivity was detected in live *M. leprae*, heat-killed bacteria also showed weak signal (S4 Fig), suggesting a potential nonspecific binding of TAG on the surface of *M. leprae*. We therefore mixed live *M. leprae* with lipid-rich cell lysate prepared from PMA-treated THP-1 cells. In this case, however, the $^{14}$C signal was not detected in the TAG fraction prepared from *M. leprae* (Fig 3G), suggesting that the nonspecific binding of TAG on live *M. leprae* is insignificant. Radioactive signals from other lipids were also detected in a darker image, suggesting the utilization of [$^{14}$C] stearic acid for the biosynthesis of other lipids (data not shown). Together, these results suggest that

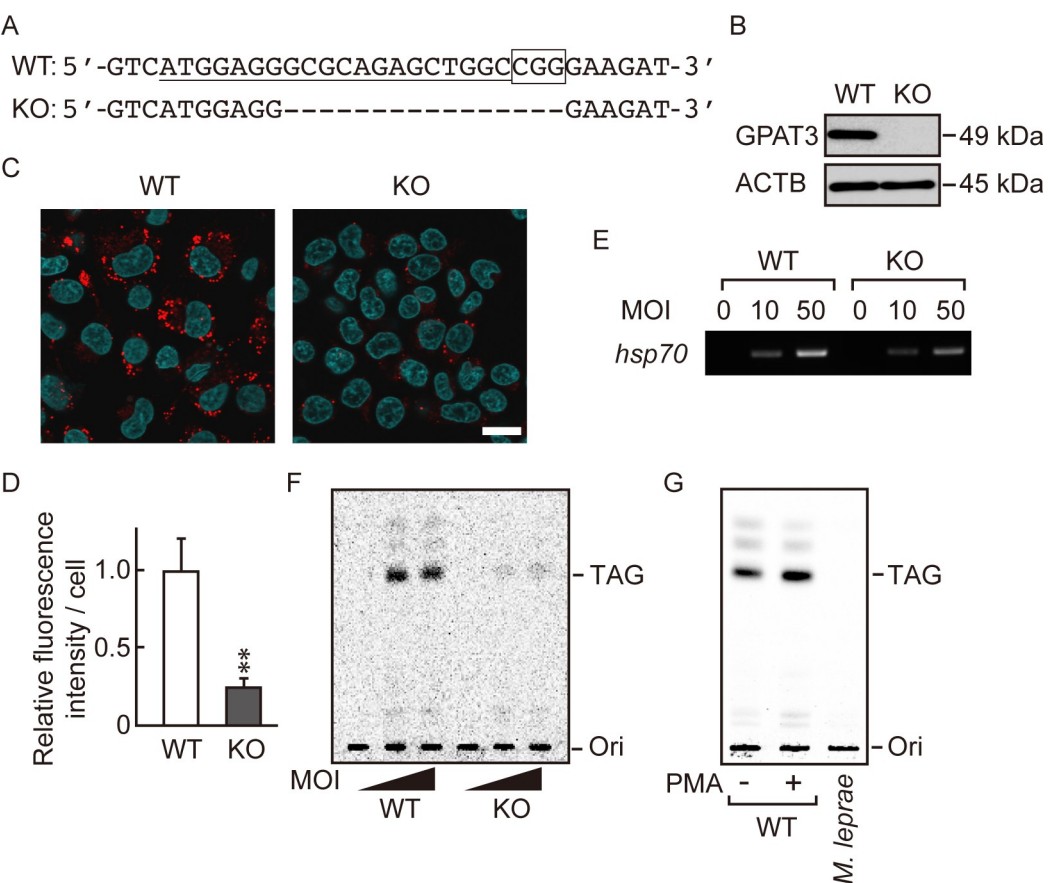

**Fig 3.** *M. leprae* **utilizes host TAG synthesized via a GPAT3-dependent pathway.** (A) The 20-bp target sequence of gRNA used for the CRISPR/Cas9 gene editing system (underlined) and the PAM sequence (boxed) in wild-type (WT) GPAT3. A dashed line in the *GPAT3* KO sequence indicates the frameshifting deletion as detected by DNA sequencing. (B) Western blot confirmed the absence of GPAT3 protein in KO cells. (C) LipidTOX staining (red) of WT and *GPAT3* KO THP-1 cells infected or uninfected with *M. leprae* (MOI: 50) for 24 h with Hoechst 33258 counterstaining (blue). Scale bar: 5 µm. (D) Quantification of LipidTOX staining using fluorescence intensity values to quantify lipid droplets. Significance was determined with a Student's *t* test. Two asterisks indicate $p < 0.01$. (E and F) WT and *GPAT3* KO THP-1 cells were cultured in medium containing 0.2 µCi of [$^{14}$C] stearic acid for 16 h after *M. leprae* infection (MOI: 10 and 50). (E) *M. leprae* isolated from infected THP-1 cells was confirmed by PCR amplification of the *M. leprae hsp70* DNA. (F) *M. leprae* was purified from cells, and the bacilli lipids were extracted and separated by TLC. (G) THP-1 cells were treated with PMA (20 ng/mL) for 24 h to promote lipid droplet formation, then incubated with 0.2 µCi of [$^{14}$C] stearic acid for 16 h. Cell lysate was sonicated and mixed with *M. leprae* then incubated for 24 h. *M. leprae* was isolated and extracted lipids were separated by TLC to evaluate radioactivity.

*M. leprae* induces *de novo* synthesis of TAG in host macrophages in a GPAT3-dependent manner, and that newly synthesized TAG following *M. leprae* infection may be utilized by *M. leprae* itself.

## GPAT3 is essential for the intracellular survival of *M. leprae*

We visualized both intracellular localization of *M. leprae* and lipid droplets by fluorescence imaging using confocal laser scanning microscopy (Fig 4A). The bacilli were counted in 30 cells and are presented as the mean and standard deviation for wild-type and *GPAT3* KO cells (Fig 4B). The MOI 200 was used to better visualize the results. After infection, *M. leprae* localized in lipid droplets in wild-type cells, whereas the number of both intracellular *M. leprae* and lipid droplets was clearly reduced in *GPAT3* KO cells.

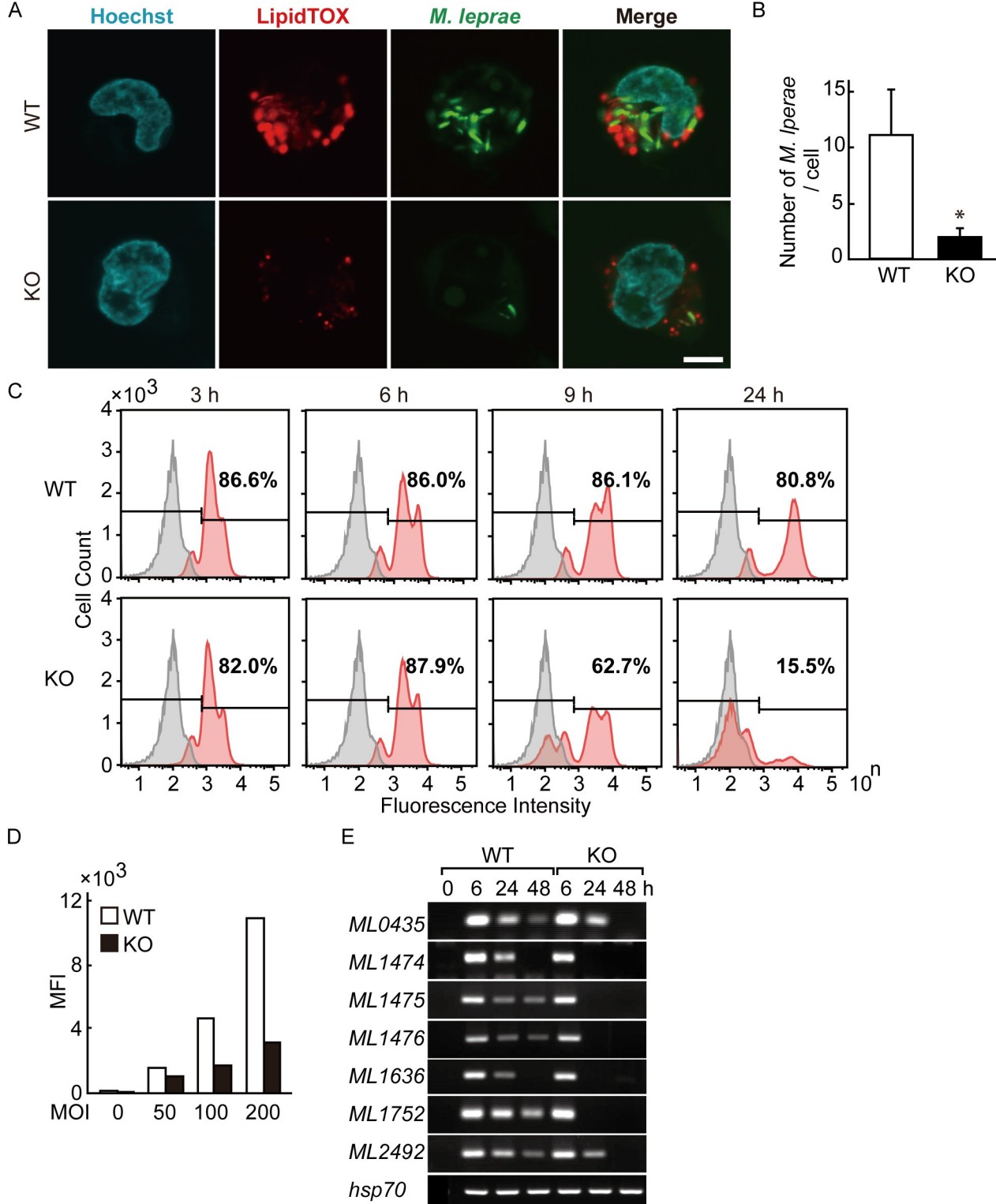

**Fig 4. Intracellular *M. leprae* viability requires GPAT3.** (A) Wild-type (WT) and *GPAT3* KO THP-1 cells infected with FITC-conjugated *M. leprae* (MOI: 200) were cultured for 24 h and confocal laser scanning microscopy was used to analyze *M. leprae* and lipid droplet localization. Fluorescence images of lipid droplets stained with LipidTOX (red), *M. leprae* (green), Hoechst 33258 (blue) and their merged images are shown. Scale bar: 5 μm. (B) Quantification of *M. leprae* in WT and *GPAT3* KO cells (n = 30 each) shown by the mean ± S.D. Significance was determined with a Student's *t* test. One asterisk indicates *p*<0.01. (C) WT and *GPAT3* KO THP-1 cells were infected with FITC-conjugated *M. leprae* (MOI: 200) for 3, 6, 9, and 24

h, and the fluorescence intensity of 100,000 cells was analyzed by flow cytometry. Gray and red peaks indicate the fluorescence intensity of uninfected and infected cells, respectively. (D) Mean fluorescence intensity (MFI) of cells infected with different MOI (0, 50, 100, and 200). (E) Wild-type and *GPAT3* KO cells ($3 \times 10^6$) were cultured in 6-well plates and inoculated with *M. leprae*. After incubation for the indicated period of time, total RNA was purified and an RT-PCR analysis for *M. leprae* pseudogenes were performed. A representative result from three independent experiments is shown in each panel.

We also performed a flowcytometric analysis of THP-1 cells to compare the mass engulfment of FITC-labeled *M. leprae* by wild-type and *GPAT3* KO cells. More than 80% of THP-1 cells were fluorescent at 3 h post-infection and the levels of such cells were similar between wild-type and *GPAT3* KO cells until 6 h (Fig 4C). The FITC-positive cells began to decline at 9 h in *GPAT3* KO cells and were reduced to 15.5% by 24 h. Conversely, the fluorescence of wild-type cells remained high (>80%) at 24 h (Fig 4C). Similarly, the mean fluorescence intensity (MFI) in each cell type was higher in wild-type cells than in *GPAT3* KO cells (Fig 4D), which is consistent with the confocal microscopy observations (Fig 4A and 4B).

We previously reported that pseudogenes and noncoding regions in live *M. leprae* are not silent but instead are strongly transcribed and can be detected as RNA [14, 18, 24, 25]. The expression levels of these RNA are variable among patients with lepromatous leprosy [18], and decrease after antibiotic treatment (unpublished data). Therefore, we used the expression levels of these RNA to estimate the viability of *M. leprae*. For seven *M. leprae* pseudogenes, similar levels of mRNA were induced by wild-type and *GPAT3* KO cells 6 h after infection. However, for *GPAT3* KO cells the *M. leprae* pseudogene expression was rapidly reduced compared to that seen for wild-type cells (Fig 4E). The expression of *M. leprae hsp70* mRNA was used as a control, since its expression level is stable [26] and did not change in both cells. These results suggest that the viability of *M. leprae* was lower in *GPAT3* KO cells. Together, these data indicate that GPAT3 expression in host cells could be important to maintain the intracellular environment required for *M. leprae* to successfully parasitize within host macrophages.

## Discussion

Our group and others have shown that *M. leprae* modifies host lipid metabolism to induce lipid droplet formation in infected host macrophages [6, 7, 27]. However, changes in lipid composition in infected cells, the mechanism associated with these changes, and the role of such lipids in *M. leprae* are not well understood. In this study, we performed a qualitative lipid analysis using HPTLC to show that TAG is the main lipid that accumulates in lipid droplets in *M. leprae*-infected THP-1 cells. TAG is known to be a major component of lipid droplets and plays a central role in maintaining energy homeostasis [28]. Sustained accumulation of intracellular TAG was unlikely to be a non-specific cell response toward bacterial components or due to macrophage activation or phagocytosis, but instead is a specific event associated with infection by viable *M. leprae*. This phenomenon involving live bacilli is similar to our previous findings [6, 7, 29].

For lipid accumulation associated with live *M. leprae*, induction of ADRP and perilipin expression is needed to enhance storage of neutral lipids and reduce the expression of HSL, which is involved in lipid droplet degradation [6, 7]. In addition, tryptophan aspartate-containing coat protein (CORO1A), which inhibits phagosome-lysosome fusion, localizes in the membrane of phagosomes containing *M. leprae* in skin lesions from leprosy patients [13, 30]. Accumulation of CORO1A persisted on the phagosome membrane around live bacilli and there was transient accumulation in dead bacilli when *M. leprae* or *M. bovis* BCG was infected in macrophages [30, 31]. Since the induction of GPAT3 expression and TAG accumulation were not observed with the addition of PGN, a major cell wall component of mycobacteria, a

hitherto unknown component(s) specific to live *M. leprae* could have significant effects on host cells. The effect of other mycobacterial species on GPAT3 expression and the accumulation of TAG in host cells should be a topic of future investigation.

TAG biosynthesis occurs mainly via the glycerol-3-phosphate (G3P) *de novo* pathway in which three fatty acid acyl chains are successively bound to a glycerol backbone (*i.e.*, acylation) [32, 33]. In the first step of acylation catalyzed by glycerol-3-phosphate acyltransferase (GPAT), an acyl-CoA is bound to G3P to form lysophosphatidic acid (LPA). Then, 1-acylglycerol-3-phosphate acyltransferase (AGPAT) catalyzes LPA to form phosphatidic acid (PA). The slow reaction rate of GPAT suggests that it is the rate-limiting enzyme in TAG biosynthesis [34].

Four different genes encode GPAT isoforms 1–4, and isoform-specific differences in tissue expression patterns, subcellular location, fatty acyl-CoA substrate preference, and sensitivity to N-ethylmaleimide have been observed [35]. In this study we demonstrated that GPAT3 expression alone was increased after *M. leprae* infection, and this increase occurred in parallel with TAG accumulation in host cells. In *GPAT3* KO cells, the formation of lipid droplets was significantly suppressed relative to wild-type cells after *M. leprae* infection. In addition, expression levels of *M. leprae* pseudogenes were rapidly reduced in *GPAT3* KO cells compared to wild-type cells, indicating that *M. leprae* viability is not sustained in *GPAT3* KO cells. Together, these results suggest a possibility that host GPAT3-mediated TAG synthesis could be responsible for the foamy-cell formation induced by *M. leprae* infection and that GPAT3 activity might be important for maintaining intracellular parasitization by *M. leprae*. Further studies are needed to confirm these points.

Other studies have reported that *M. leprae* infection promotes cholesterol accumulation rather than TAG [8]. This discrepancy may be due to the different MOI and cell systems used. The number of bacteria used in our study was 10 times higher than that in previous reports. Additionally, the cells used in the previous report were primary cultures of human macrophages, and primary cells infected with *M. leprae* are known to be rich in cholesterol and cholesterol esters [27]. Differences in the lipid moieties accumulated in response to *M. leprae* infection between primary cells and THP-1 cells need to be clarified in the future. Two additional bands can be seen above the TAG spot in wild-type cells (Fig 3F) that could be metabolites of the TAG synthesis pathway, but the nature of these moieties is not yet clear.

How *M. leprae* infection induces host GPAT3 expression is not clear. Activation of the peroxisome proliferator-activated receptor γ (PPARγ) signaling pathway is reported to be responsible for up-regulation of *Gpat3* gene expression during adipocyte differentiation [20, 36, 37]. PPARγ is an important transcription factor that regulates expression of genes that are closely related to lipogenesis, lipid metabolism, and foam cell formation in macrophages [38]. In addition to GPAT3, PPARγ also targets ADRP [39], which has a similar induction following *M. leprae* infection in THP-1 cells [6]. Recently, we have reported that activation of PPARγ and PPARδ is important for lipid accumulation in *M. leprae*-infected THP-1 cells [40]. In Schwann cells, phenolic glicolipid-1 (PGL-1) of *M. leprae* promoted lipid droplet formation by activating crosstalk between CD206 and PPARγ [41]. Therefore, *M. leprae* might utilize the signal transduction pathway(s) mediated by PPARγ to induce GPAT3 expression in infected cells.

We showed that intracellular localization of *M. leprae* to lipid droplets was clearly reduced in *GPAT3* KO cells compared to wild-type cells. This observation is in agreement with a recent study showing that in bone marrow-derived macrophages from *Gpat3*$^{-/-}$ mice, not only was the formation of Kdo$_2$-lipid A (KLA)-inducible lipid droplets suppressed, intriguingly, the phagocytic capacity of the macrophages was also reduced [42]. In the present study, however, we have clearly shown that GPAT3 does not affect the ability of THP-1 cells to undergo phagocytosis. The role of GPAT3 on the phagocytosis of different cell types should be elucidated in the future.

Another interesting question is how *M. leprae* utilizes intracellular lipids during its parasitism. Lipids represent approximately 60% of the dry weight of mycobacteria cell walls and include lipid varieties such as phospholipids, glycolipids and mycolic acid (a long fatty acid). An intact "greasy" cell wall is considered to be important for mycobacteria viability. Medications such as isoniazid, ethionamide, isoxyl and thiacetazone that inhibit mycolic acid synthesis have shown antimycobacterial effects [43, 44]. From this perspective, *M. leprae* may rely on host-derived lipids to generate its own cell wall. Here, we labeled newly-synthesized intracellular TAG with [14C] stearic acid to examine incorporation of host-derived lipids in the cell wall of *M. leprae* following infection. The labeled TAG content of intracellular *M. leprae* in wild-type cells was markedly higher than that in *GPAT3* KO cells. These results suggest a possibility that *M. leprae* utilizes host-derived lipids during intracellular parasitization.

*M. leprae* has substantially fewer functional genes compared to *M. tuberculosis* [45]. Several of the lost genes are involved in lipid metabolism and the lack of these genes could be associated with their slow growth *in vivo* and the inability to cultivate *M. leprae in vitro*. However, across evolution, *M. leprae* acquired the ability to use host cell resources to support its parasitism. In this study, we showed that *M. leprae* infection increased intracellular TAG accumulation via induction of host GPAT3 expression that in turn maintains a lipid-rich environment in host macrophages. Our finding indicates that GPAT3 also plays an important role in the intracellular survival of *M. leprae*, suggesting that GPAT3 may become a novel target for leprosy treatment.

## Supporting information

**S1 Table. List of primers used in RT-PCR.**
(DOCX)

**S1 Fig. Confirmation of the lipid fraction in *M. leprae*-infected THP-1 cells using standard controls.** THP-1 cells ($3 \times 10^6$) were cultured in 6-well plates with live *M. leprae* (MOI: 20) for 24 h. Total lipids extracted from cells with 20 nmol of control lipids (PL, Cho, FA and TAG) were spotted on an HPTLC plate. After separation, the plate was stained with a charring solution containing 10% $CuSO_4$ and 8% $H_3PO_4$ and heated at 180°C for 10 min.
(TIF)

**S2 Fig. The endogenous expression levels of GPAT isoforms in THP-1 cells.** Total RNA was extracted and the expression level of GPAT isoforms was evaluated by qRT-PCR. The results were normalized relative to *ACTB* levels. Each bar represents the mean ± S.D in triplicate.
(TIF)

**S3 Fig. *M. leprae* infection does not affect GPAT4 expression in GPAT3 KO cells.** WT and *GPAT3* KO cells ($3 \times 10^6$) were cultured in 6-well plate and infected with live *M. leprae* (MOI: 50). After incubating for the indicated time, total RNA was purified and RT-PCR analysis was performed.
(TIF)

**S4 Fig. Analysis of nonspecific binding of TAG on the surface of *M. leprae*.** Wild-type THP-1 cells were inoculated with either live or heat-killed *M. leprae* (MOI: 10 and 50), then cultured with 0.2 μCi of [14C] stearic acid for 16 h. *M. leprae* was isolated and extracted lipids were separated by TLC to evaluate radioactivity.
(TIF)

**S1 Raw images.**
(TIF)

**S2 Raw images.**
(TIF)

## Acknowledgments

We are grateful to Prof. J. Aoki and Dr. A. Inoue (Tohoku University, Sendai, Japan) for valuable technical advice regarding the CRISPR-Cas9 system.

## Author Contributions

**Conceptualization:** Kazunari Tanigawa, Yasuhiro Hayashi, Kotaro Hama, Atsushi Yamashita, Ken Karasawa, Koichi Suzuki.

**Data curation:** Kazunari Tanigawa.

**Formal analysis:** Kazunari Tanigawa.

**Funding acquisition:** Kazunari Tanigawa, Koichi Suzuki.

**Investigation:** Kazunari Tanigawa.

**Methodology:** Kazunari Tanigawa, Yasuhiro Hayashi, Kotaro Hama, Atsushi Yamashita, Yuqian Luo, Koichi Suzuki.

**Project administration:** Ken Karasawa, Koichi Suzuki.

**Resources:** Kazunari Tanigawa, Yasuhiro Hayashi, Kotaro Hama, Atsushi Yamashita, Kazuaki Yokoyama, Yuqian Luo, Akira Kawashima, Yumi Maeda, Yasuhiro Nakamura, Ayako Harada, Mitsuo Kiriya, Ken Karasawa, Koichi Suzuki.

**Software:** Kazunari Tanigawa, Yasuhiro Hayashi.

**Supervision:** Koichi Suzuki.

**Validation:** Kazunari Tanigawa.

**Visualization:** Kazunari Tanigawa.

**Writing – original draft:** Kazunari Tanigawa.

**Writing – review & editing:** Kazunari Tanigawa, Yasuhiro Hayashi, Yuqian Luo, Mitsuo Kiriya, Koichi Suzuki.

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
