## [Decision Letter · Decision Letter 0]

10 Sep 2020

PONE-D-20-21795

Mycobacterium leprae promotes triacylglycerol de novo synthesis through induction of GPAT3 expression in host macrophages

PLOS ONE

Dear Dr. Suzuki,

Thank you for submitting your manuscript to PLOS ONE. After careful consideration, we feel that it has merit but does not fully meet PLOS ONE’s publication criteria as it currently stands. Therefore, we invite you to submit a revised version of the manuscript that addresses the points raised during the review proce

Both  reviewers  did not agree with your conclusions and  felt there were serious flaws that need to be resurrected.  This also entails new experiments.  

Please submit your revised manuscript within 90days. If you will need more time than this to complete your revisions, please reply to this message or contact the journal office at plosone@plos.org. Please include the following items when submitting your revised manuscript:

We look forward to receiving your revised manuscript.

Kind regards,

Delphi Chatterjee

Academic Editor

PLOS ONE

Journal Requirements:

2.PLOS ONE now requires that authors provide the original uncropped and unadjusted images underlying all blot or gel results reported in a submission’s figures or Supporting Information files. This policy and the journal’s other requirements for blot/gel reporting and figure preparation are described in detail at https://journals.plos.org/plosone/s/figures#loc-blot-and-gel-reporting-requirements and https://journals.plos.org/plosone/s/figures#loc-preparing-figures-from-image-files. When you submit your revised manuscript, please ensure that your figures adhere fully to these guidelines and provide the original underlying images for all blot or gel data reported in your submission. See the following link for instructions on providing the original image data: https://journals.plos.org/plosone/s/figures#loc-original-images-for-blots-and-gels.

Reviewers' comments:

Reviewer's Responses to Questions

**Comments to the Author**

1. Is the manuscript technically sound, and do the data support the conclusions?

Reviewer #1: Partly

Reviewer #2: Partly

2. Has the statistical analysis been performed appropriately and rigorously? 

Reviewer #1: Yes

Reviewer #2: Yes

3. Have the authors made all data underlying the findings in their manuscript fully available?

Reviewer #1: Yes

Reviewer #2: Yes

4. Is the manuscript presented in an intelligible fashion and written in standard English?

Reviewer #1: Yes

Reviewer #2: Yes

5. Review Comments to the Author

Reviewer #1: The manuscript entitled ‘Mycobacterium leprae promotes triacylglycerol de novo synsthesis through induction of GPAT3 expression in host macrophages’ by Tanigawa et al; claims to demonstrate that M. leprae infection induces the production of triacylglycerols (TAGs) in a macrophage cell line. Authors linked this host production of TAGs to the function of glycerol-3-phosphagte acyltransferase 3 (GPAT3), which may be essential for intracellular survival of M. leprae within host cells. This is an interesting study working with a difficult pathogen, defining how M. leprae could stimulate the host cell physiology to create an environment that favors its intracellular survival. This study is well conducted and sounded; however, authors have some bold claims that are not well supported by their data. Major concerns are:

1/ Analysis of host lipids performing HPTLC. Although this is a valid technique, a LC/MS analysis will provide a clear picture (and quantification) of all the host lipids that are being altered after M. leprae infection of TH-1 cells.

As depicted, authors focused on DAGs/TAGs and the phospholipid fractions (a major fraction in host lipids) is not clearly studied in the solvent system presented (PLs practically did not migrated on the HPTLC from the origin to state that these are not being modified). Moreover, it will be important to add reference standard controls on the HPTLCs to be certain of the nature of the lipids being identified. THP-1 cells lipids and M. leprae lipids alone controls are also missing. Provide also the TLCs in color, so this can be better appreciate it.

2/ In the production of TAGs by host cells it also seems to be a dose dependent effect (different MOIs induced different expression levels at 24 h), but this is not being follow up. Authors, did also not explain why there is a spike of TAG production in THP-1 cells infected with dead M. leprae.

3/ Related to the induction of GPAT3, it will be important to assess if this is induction is M. leprae specific or if other bacteria/mycobacteria induces GPAT3 in Th-1 cell line. For example, M. bovis BCG, M. avium, M. smegmatis, or even M. tuberculosis.

4/ Fig 3C needs better images and with the same background contrast. GPAT3 KO background seems lighter than the WT background making difficult to interpret. A quantification will be also required demonstrating the author’s claim of decrease in lipid formation in GPAT3 cells infected with live M. leprae.

5/ Line 291-292: ‘newly synthesized TAG following M. leprae infection is in turn utilized by M. leprae itself’ There is not conclusive data supporting this statement. Could authors rationalize why other M. leprae lipids are not radioactive and thus not being seeing on the autoradiogram provided? One will be expecting seeing radioactive free fatty acids, DAG, etc. and not only DAG.

6/ Fig. 4A: Quantification by Confocal microscopy counting bacteria per cell is necessary. It will be important to show the flow cytometry plots describing the strategy used removing extracellular FITC-M. leprae bacilli and focusing only in infected cells. Also to show graphs at % of infected vs. non-infected cells. How many times this study was done?

7/ A good control will be to have one of the others GPAT KOs, and further determine that their absence does not affect M. leprae uptake/survival in THP-1 cells.

8/ Lines 379-381: “Together, these results suggest that host GPAT-3-mediated TAG synthesis is responsible for foamy cell formation induced by M. leprae infection, and that GPAT3 activity is necessary for maintaining intracellular parasitization by M. leprae.”This is not proven in this manuscript. The infection went in vitro for 24 h. It may participate in the establishment of the infection in THP-1 cells.

9/ Line 415-416: It is not show that the produced host TAG can be used as a source for mycobacterial lipids and that GPAT3 is essential (would say plays an important role) for intracellular survival of M. leprae.

10/ Discussion: Is the absence of GPAT3 affecting M. leprae replication, survival or both?

11/ Fig 2A does not show what is discussed in the text.

12/ In Fig 2B: It is not clear if the statistical analyses are relative to ACTB, to others GPAT measured, or if these are comparing GPAT3 RNA transcription levels across time. In figure legends, it is not clear how many times these analyses were performed.

13/ Some figure legends have a Scale bar that do not apply.

14/ Animals studies are missing but an ethics statement is provided.

Reviewer #2: General comments

This article explores the mechanisms by which M. leprae modulates host cell lipid metabolism. The capacity of M. leprae to induce lipid accumulation in infected cells, an aspect that seems to be essential for bacterial pathogenesis, has been demonstrated by several reports during the last two decades. In this study, the authors infected cells of the monocytic cell line THP-1 and based on this model of infection, they concluded that TAG are the major class of lipids accumulated during infection. They also show that M. leprae induces the expression of GPAT3, one of the four isoforms of the enzyme that catalyzes the rate-limiting step in the pathway of TAG biosynthesis. Moreover, by Knocking down the GPAT3 gene in THP-1 cells, they conclude that host TAG is used by M. leprae and that this nutrient source is important for bacterial intracellular survival. Although the subject is relevant and the data generated are original, experiments are incomplete, controls are missing, and alternative methods are needed to validate the conclusions of the study. Importantly, the authors should consider the limitations imposed by their in vitro model based on THP-1 cells and discuss the discrepancies observed between their results and those generated by others along recent years.

Specific comments

Title

In the title it is said that the observations are in infected macrophages, however, this is not thru since no protocol for differentiating THP-1 monocytes into macrophages is mentioned in the methods. This needs to be corrected.

Results

Figure 1- Results differ from Mattos et al. 2014, in which in human primary monocytes/ macrophages infected with M. leprae in a MOI of 5:1 cholesterol and cholesterol esters are the most abundant lipids and TAG actually decreases with the infection. The cell model is different, and MOI is higher in this work (50:1). This needs to be addressed in the discussion section. Also, tumor cells frequently express a lipogenic phenotype and how this may influence the results observed in THP-1 monocytes should be addressed at some point.

Figure 2- From the literature it seems that GPAT1 and 2 are located in the mitochondria, while GPAT3 and 4 are related to LDs. It would be interesting to show the relative basal level of expression of each isoform in THP-1 monocytes. Is GPAT3 the more abundant isoform in this cell line?

Figure 3- It would be interesting to show that KO cells of the other GPAT isoforms, specially GPAT4, are not related to TAG biosynthesis in the context of M. leprae infection. This would strengthen the data from the previous figure that only showed modulation of GPAT3 expression. 3C- The quality of the microscopy images is not good. I would also suggest increasing the size. 3E- The conclusion drawn from this experiment that M. leprae utilizes host TAG is not convincing. During cell disruption and M. leprae isolation, TAG molecules could nonspecifically bind to the hydrophobic bacterial cell envelope. This would be more likely to occur in WT cells where the levels of labeled TAG are higher. To rule out this possibility, the inclusion of a control in which heat-killed M. leprae is incubated with cell lysates prepared from monocytes pre-labeled for 24 h with stearic C14 is strongly recommended. Also, based on the results shown in Figure 4, it would be expected a smaller number of bacilli being recovered from KO cells. So, why the signal of the hsp70 gene is similar in WT and KO cells? Finally, a second TLC showing the levels of labeled TAG in the host cell fraction will nicely complement the data. There are two slight bands above TAG in the WT infected cells that are also not seen in the GPAT3 KO. Would the authors believe that these are products derived of TAG utilization by M. leprae or other possible products related to GPAT3?

Figure 4- Figures A-C show that GPAT3 KO is apparently affecting M. leprae internalization, since less bacteria is seen inside the cells at 24 h of infection. The reduced number of bacteria in KO cells could also be related to a decrease in mycobacterial viability. So, the inclusion of earlier time points of infection in the analysis shown in figures A-C, such as 4-6 h, is imperative and will definitively discriminate between these possibilities. If KO cells show less phagocytic capacity, as described in an earlier study referred in the discussion, this will absolutely compromise the conclusions drawn from figures 3 and 4. The reduction in LDs accumulation in the KO cells could be because less bacteria is infecting the cell and therefore less modulation of lipid metabolism is seen and not so much because of the absence of GPAT3. Concerning figure D, a detailed explanation of how bacterial viability is determined is missing in the methods section. In this case, different from figure 3, they mention that mRNA instead of DNA of the hsp70 gene was measured. Why? In these experiments they are also using a MOI of 200. Why? I couldn’t follow whether hsp70 was used to normalize potential differences in bacterial loads between WT and KO cells. I would advise to do a qRT-PCR, which is indeed more quantitative to evaluate viability percentages. Actually, in several other studies M. leprae viability has been determined by the 16S rRNA/16S rDNA ratio as described by Martinez AN et al, J Clin Microbiol. 2009; 47(7):2124–30.

Discussion

This section will need an extensive revision after the inclusion of new experiments as pointed out before. So far, the conclusions taken based on the assays with the GPAT-3 KO cells are not correct. The authors should also take into account the limitations imposed by their in vitro model based on THP-1 cells and discuss the discrepancies observed between their results and those generated by others along recent years. Previous studies have shown that in context of human primary monocytes, both live and dead M. leprae induces LDs accumulation. Also, infected primary monocytes were found enriched in cholesterol and cholesterol ester LDs, finding not confirmed in THP-1 cells.

6. PLOS authors have the option to publish the peer review history of their article (what does this mean?). If published, this will include your full peer review and any attached files.

Reviewer #1: No

Reviewer #2: No

---

## [Author Response · Author response to Decision Letter 0]

13 Jan 2021

Reviewer I

The manuscript entitled ‘Mycobacterium leprae promotes triacylglycerol de novo synsthesis through induction of GPAT3 expression in host macrophages’ by Tanigawa et al; claims to demonstrate that M. leprae infection induces the production of triacylglycerols (TAGs) in a macrophage cell line. Authors linked this host production of TAGs to the function of glycerol-3-phosphagte acyltransferase 3 (GPAT3), which may be essential for intracellular survival of M. leprae within host cells. This is an interesting study working with a difficult pathogen, defining how M. leprae could stimulate the host cell physiology to create an environment that favors its intracellular survival. This study is well conducted and sounded; however, authors have some bold claims that are not well supported by their data.

Response: 

We thank the reviewer for the comments, especially his/her noting that “This is an interesting study working with a difficult pathogen, defining how M. leprae could stimulate the host cell physiology to create an environment that favors its intracellular survival. This study is well conducted and sounded.”

We have addressed reviewer’s concerns below.

1-1. Analysis of host lipids performing HPTLC. Although this is a valid technique, a LC/MS analysis will provide a clear picture (and quantification) of all the host lipids that are being altered after M. leprae infection of THP-1 cells. 

Response: 

We agree with the reviewer that an LC/MS analysis would provide clearer results for assessing the changes in intracellular lipid composition within M. leprae-infected macrophages. In this study, we have analyzed cellular lipids using the HPTLC method that was previously employed by Mattos et al. (Mattos et al., Cell Microbial 16: 797-815, 2014). In a separate set of studies, we have performed an in-depth evaluation of the molecular species of accumulated TAG by LC-MS/MS analysis as shown below (Figure A). However, the results will be reported in the future, as it is still ongoing.

1-2. As depicted, authors focused on DAGs/TAGs and the phospholipid fractions (a major fraction in host lipids) is not clearly studied in the solvent system presented (PLs practically did not migrated on the HPTLC from the origin to state that these are not being modified). Moreover, it will be important to add reference standard controls on the HPTLCs to be certain of the nature of the lipids being identified. THP-1 cells lipids and M. leprae lipids alone controls are also missing. Provide also the TLCs in color, so this can be better appreciate it.

Response:

As the reviewer pointed out, the separation of PL is difficult in the solvent system we used. However, the purpose of this experiment is to evaluate the major lipid that accumulates in the lipid droplets of M. leprae-infected THP-1 cells. Since the main components of the lipid droplets are neutral lipids, we performed experiments with a solvent system that is suitable for separating TAG and Cho. We have described this issue in the Results section as follows (page 11, lines 222-224):

“To clarify the effect of an M. leprae infection on host cell lipid composition, we first examined the lipid components of M. leprae-infected THP-1 cells using HPTLC conditions suitable for separating neutral lipids [8].”

According to the Reviewer’s suggestion, we added standard controls to TLC analysis that show clear separation of different lipid components (S1 Fig). Also, in HPTLC, it is known that the RF (relative to front) value in this solvent system can be used to estimate the obtained lipid (Mattos et al., Cell Microbial 16: 797-815, 2014). To explain this, we have modified the sentence in the Results section as “The positions of cholesterol ester (ChoE), triacylglycerol (TAG), fatty acid (FA), cholesterol (Cho), diacylglycerol (DAG), monoacylglycerol (MAG) and phospholipid (PL) were determined by their relative to front (RF) values based on the position of standard lipids (S1 Fig).” (page 11, lines 225-228). 

We have also provided data from the HPTLC analysis of M. leprae and THP-1 cell alone controls in Fig. 1B. TAG was not detected in M. leprae, suggesting that the increased TAG post-infection was a host-derived lipid. We briefly discussed this in our revised manuscript as “Since TAG was not detected in the lipids derived from M. leprae before infection (Fig 1B), it suggests that the observed lipids were derived from the host cells.” (page 11, lines 237-238).

Regarding the color of the TLC, only a single color (gray) is visualized in a different density in our system, since copper sulphate is sprayed on to the plate following heating to detect neutral lipids. Therefore, we presented the results in as a gray scale image. We hope for the reviewer’s understanding of the situation.

2. In the production of TAGs by host cells it also seems to be a dose dependent effect (different MOIs induced different expression levels at 24 h), but this is not being follow up. Authors, did also not explain why there is a spike of TAG production in THP-1 cells infected with dead M. leprae.

Response: 

As the Reviewer pointed out, the effects of M. leprae on THP-1 cells are dose-dependent as well as time-dependent until 48 h in our study. In addition, heat-killed M. leprae showed a transient effect (or spike) around 6 h after stimulation but returned to its original level by 24 h, as the reviewer noted. However, these are not novel observations as we have reported them previously following addition of live or heat-killed M. leprae (Tanigawa et al., FEMS Microbial Lett 289: 72-79, 2008, Tanigawa et al., Microb Pathog 52: 285-291, 2012.). The transient effect of heat-killed M. leprae on host TAG production was also observed following the addition of latex beads, although that was not evident in the present study. To clarify these points, we have rephrased the sentence in the Results as follows (page 11, lines 230-234): 

 “TAG accumulation was transiently induced by dead bacilli at 6 h, but the levels returned to baseline within 24 h (Fig 1A). The observed transient TAG induction corresponds with previous observations that heat-killed M. leprae changed host mRNA levels of ADRP, perilipin and HSL [6, 7].”

3. Related to the induction of GPAT3, it will be important to assess if this is induction is M. leprae specific or if other bacteria/mycobacteria induces GPAT3 in Th-1 cell line. For example, M. bovis BCG, M. avium, M. smegmatis, or even M. tuberculosis.

Response: 

We agree with the reviewer’s comment. It will be of interest to examine the changes in GPAT3 mRNA expression in regard to intracellular lipid metabolism. However, we have been studying M. leprae for its effect on the accumulation of large amounts of lipids within the phagosomes of infected macrophage, because such lipid accumulation is not observed in other bacterial infections and is specific to lepromatous type leprosy. In addition, PGN, a rich cell wall component common to mycobacteria, did not induce TAG production. Therefore, we consider TAG production to be caused by an unknown component(s) derived from live M. leprae. We agree with the reviewer that studying the potential effects of other mycobacteria on the expression of GPAT3 and accumulation of TAG in host cells is an interesting subject to be explored in the future. 

We briefly discussed this in the Discussion of our revised manuscript as follows (pages 19-20, lines 422-427):

“Since the induction of GPAT3 expression and TAG accumulation were not observed with the addition of PGN, a major cell wall component of mycobacteria, a hitherto unknown component(s) specific to live M. leprae could have significant effects on host cells. The effect of other mycobacterial species on GPAT3 expression and accumulation of TAG in host cells should be a topic of future investigation.”

4. Fig 3C needs better images and with the same background contrast. GPAT3 KO background seems lighter than the WT background making difficult to interpret. A quantification will be also required demonstrating the author’s claim of decrease in lipid formation in GPAT3 cells infected with live M. leprae.

Response:

We have performed fluorescent staining using the LipidTOX reagent to visualize cellular lipid droplets to replace the Oil red O staining shown in Fig 3C. The results show a clear difference in the lipid droplet formation between Wild-type and GPAT3 KO cells. The brightness of each cell was evaluated using the ROI (region of interest) analysis with Olympus FV10i software and illustrated in Fig 3D. Quantitative analysis showed a reduced accumulation of lipid droplets in GPAT3 KO cells following M. leprae infection. 

The Results sections was amended as follows (page 14, lines 304-309). 

“We then compared the effect of M. leprae infection on lipid droplet formation in wild-type and GPAT3 KO cells using LipidTOX staining (Fig 3C) and quantified the LipidTOX fluorescence intensity in cells (Fig 3D). We observed a large number of lipid droplets in wild-type cells following M. leprae infection, whereas the number of lipid droplets was reduced by about 80% in GPAT3 KO cells, compared to the wild-type cells.”

The Materials and Methods and figure legend were corrected accordingly (page 8, lines 175-177 and pages 15, lines 338-340).

5. Line 291-292: ‘newly synthesized TAG following M. leprae infection is in turn utilized by M. leprae itself’ There is not conclusive data supporting this statement. Could authors rationalize why other M. leprae lipids are not radioactive and thus not being seeing on the autoradiogram provided? One will be expecting seeing radioactive free fatty acids, DAG, etc. and not only DAG.

Response: 

Based on the reviewer’s comment, we have rephrased the sentence as follows (page 15, lines 332-335):

“Together, these results suggest that M. leprae induces de novo synthesis of TAG in host macrophages in a GPAT3-dependent manner, and that newly synthesized TAG following M. leprae infection may be utilized by M. leprae itself.”

Regarding radioactive signals from other lipids, those signals can be detected in darker image as the reviewer pointed out. Since the TAG signal is so strong, the signal from other lipids could not be visualized at this intensity. We added this brief description to the Results section as follows (page 15, lines 327-329):

“Radioactive signals from other lipids were also detected in a darker image, suggesting the utilization of [14C] stearic acid for the biosynthesis of other lipids (data not shown).”

6. Fig. 4A: Quantification by Confocal microscopy counting bacteria per cell is necessary. It will be important to show the flow cytometry plots describing the strategy used removing extracellular FITC-M. leprae bacilli and focusing only in infected cells. Also to show graphs at % of infected vs. non-infected cells. How many times this study was done?

Response: 

According to the Reviewer’s suggestion, we counted the number of M. leprae cells within THP-1 cells. The mean numbers of M. leprae in 30 cells are illustrated in Fig 4B, which shows a clear reduction in GPAT3 KO cells. This was described in the Results section as follows (page 16, lines 353-359):

“We visualized both intracellular localization of M. leprae and lipid droplets by fluorescence imaging using confocal laser scanning microscopy (Fig 4A). The bacilli were counted in 30 cells and are presented as the mean and standard deviation for wild-type and GPAT3 KO cells (Fig 4B). The MOI 200 was used to better visualize the results. After infection, M. leprae localized in lipid droplets in wild-type cells, whereas the number of both intracellular M. leprae and lipid droplets was clearly reduced in GPAT3 KO cells.” 

We have re-examined the time-dependent changes in intracellular M. leprae using flowcytometric analysis, and the % of positive cells has been added to the figure (Fig. 4C). The method to remove extracellular M. leprae prior to flowcytometry was described in the Materials and Methods as follows (pages 8-9, lines 181-186):

“The culture medium was discarded, the cells were washed three times with 3 ml of warm PBS to remove extracellular M. leprae, and then fixed with 3% buffered formalin. Cells were suspended in PBS with 1 mM EDTA to analyze the fluorescence intensity using a FACSCanto II flow cytometer (BD Biosciences) and FlowJo software (FlowJo LLC, Ashland, OR).”

The experiments were performed three times independently, which was added to the Figure legends accordingly (page 18, lines 399-400).

7. A good control will be to have one of the others GPAT KOs, and further determine that their absence does not affect M. leprae uptake/survival in THP-1 cells.

Response: 

We thank the reviewer for pointing out this issue. Among the four isoforms, GPAT4 and GPAT3 are microsomal enzymes that share similar functions. However, as shown in Fig. 2, the only isoform induced by M. leprae infection was GPAT3. Furthermore, we have confirmed that the expression of GPAT4 in GPAT3 KO cells did not change even after M. leprae infection (S3 Fig). From these results, we consider GPAT3 as the main isoform responding to M. leprae infection. Examination of the potential role of other GPATs in M. leprae infection could be a subject of future studies.

To explain this, we added the following sentence in the text (page 14, lines 309-312):

“Although both GPAT3 and GPAT4 are microsomal enzymes that share similar functions, the expression of GPAT4 in GPAT3 KO cells did not change, even after M. leprae infection (S3 Fig), confirming that GPAT3 is the main isoform that responds to M. leprae infection (Fig 2).” 

We thank the reviewer for their comment to improve our manuscript:

8. Lines 379-381: “Together, these results suggest that host GPAT-3-mediated TAG synthesis is responsible for foamy cell formation induced by M. leprae infection, and that GPAT3 activity is necessary for maintaining intracellular parasitization by M. leprae.” This is not proven in this manuscript. The infection went in vitro for 24 h. It may participate in the establishment of the infection in THP-1 cells.

Response: 

Based on the reviewer’s comment, we changed the sentence as follows (page 20, lines 443-447):

“Together, these results suggest a possibility that host GPAT3-mediated TAG synthesis could be responsible for the foamy-cell formation induced by M. leprae infection and that GPAT3 activity might be important for maintaining intracellular parasitization by M. leprae. Further studies are needed to confirm these points.”

9. Line 415-416: It is not show that the produced host TAG can be used as a source for mycobacterial lipids and that GPAT3 is essential (would say plays an important role) for intracellular survival of M. leprae.

Response:

According to the Reviewer’s suggestion, we removed the sentence noting that host TAG can be used as a source for mycobacterial lipids, and changed the last sentence as follows (page 22, lines 492-494):

“Our finding indicates that GPAT3 also plays an important role in the intracellular survival of M. leprae, suggesting that GPAT3 may become a novel target for leprosy treatment.”

10. Discussion: Is the absence of GPAT3 affecting M. leprae replication, survival or both?

Response: 

Our data in GPAT3 KO cells showed that the internalization of M. leprae was suppressed (Figs 4A-C), and that viability in cells seems to be reduced (Fig. 4D). We have not evaluated the replication of M. leprae since its doubling time is at least 14 days. Based on this comment, we have rephrased a sentence in the Results section as follows (page 17, lines 379-382):

“Together, these data indicate that GPAT3 expression in host cells could be important to maintain the intracellular environment required for M. leprae to successfully parasitize within host macrophages.”

11. Fig 2A does not show what is discussed in the text.

Response: 

We have revised the sentence referencing Fig 2A in the Results as follows (page 12, lines 266-267):

“Among the four isoforms, only GPAT3 mRNA levels increased after M. leprae infection at MOI 10 and 20 in 24 h (Fig 2A)”

12. In Fig 2B: It is not clear if the statistical analyses are relative to ACTB, to others GPAT measured, or if these are comparing GPAT3 RNA transcription levels across time. In figure legends, it is not clear how many times these analyses were performed.

Response: 

We thank the Reviewer for pointing out this tissue. ACTB was used to normalize mRNA levels of GPATs, and statistical analysis was performed to evaluate the changes against 0 h. The experiments were performed three times. We have completely revised the figure legend as follows:

“(B and C) THP-1 cells were cultured in 6-well plates and inoculated with M. leprae (MOI: 20). Total RNA and protein were purified from the cells for (B) qRT-PCR and (C) Western blotting analysis. GPAT mRNA levels were normalized against ACTB and are expressed relative to its level at 0 h.” (page 13, lines 288-291)

“The representative results of three independent experiments are shown.” (page 14, line 296)

13. Some figure legends have a Scale bar that do not apply.

Response: 

We have added a scale bar to Fig 3C and corrected the figure legends for Figs 3 and 4. Thank you for this comment. 

14. Animals studies are missing but an ethics statement is provided.

Response: 

The ethics statement for animal study is for the growth and preparation of M. leprae in the nude mice at Leprosy Research Center, National Institute of Infectious Diseases. We have moved ethics statement under “M. leprae preparation and cell culture” as follows (page 5, lines 89-100):

“The M. leprae were grown in the footpads of nude mice and prepared at the Leprosy Research Center, National Institute of Infectious Diseases, Tokyo, Japan as described previously [12-14]. The human premonocytic cell line THP-1 was obtained from the American Type Culture Collection (ATCC; Manassas, VA). Cells were cultured in 10-cm tissue culture dishes in RPMI-1640 medium supplemented with 10% charcoal-treated fetal bovine serum (FBS) and 50 mg/ml penicillin/streptomycin at 37�C and 5% CO2. THP-1 cells (3 × 106) were treated with latex beads (Fluoresbrite microspheres; Technochemical, Tokyo, Japan), peptidoglycan (Sigma, St Louis, MO) or live or heat-killed (80°C, 30 min) bacilli (1.5 × 108) at a multiplicity of infection (MOI) of 50. The animal experiment was reviewed and approved by the Experimental Animal Committee of the National Institute of Infectious Diseases (Permit No. 118028), and all experiments were conducted according to the recommended guidelines.”

 

Reviewer II

This article explores the mechanisms by which M. leprae modulates host cell lipid metabolism. The capacity of M. leprae to induce lipid accumulation in infected cells, an aspect that seems to be essential for bacterial pathogenesis, has been demonstrated by several reports during the last two decades. In this study, the authors infected cells of the monocytic cell line THP-1 and based on this model of infection, they concluded that TAG are the major class of lipids accumulated during infection. They also show that M. leprae induces the expression of GPAT3, one of the four isoforms of the enzyme that catalyzes the rate-limiting step in the pathway of TAG biosynthesis. Moreover, by Knocking down the GPAT3 gene in THP-1 cells, they conclude that host TAG is used by M. leprae and that this nutrient source is important for bacterial intracellular survival. Although the subject is relevant and the data generated are original, experiments are incomplete, controls are missing, and alternative methods are needed to validate the conclusions of the study. Importantly, the authors should consider the limitations imposed by their in vitro model based on THP-1 cells and discuss the discrepancies observed between their results and those generated by others along recent years.

Response:

We thank the reviewer for the comments. We have performed additional experiments, revised figures, and amended the text to address the reviewer’s concerns. Detailed point-by-point responses are summarized below.

1. Title.

In the title it is said that the observations are in infected macrophages, however, this is not thru since no protocol for differentiating THP-1 monocytes into macrophages is mentioned in the methods. This needs to be corrected. 

Response:

According to the reviewer’s suggestion, we revised “in host macrophages” to “in human premonocytic THP-1 cells” in the title.

2. Figure 1.

Results differ from Mattos et al. 2014, in which in human primary monocytes/ macrophages infected with M. leprae in a MOI of 5:1 cholesterol and cholesterol esters are the most abundant lipids and TAG actually decreases with the infection. The cell model is different, and MOI is higher in this work (50:1). This needs to be addressed in the discussion section. Also, tumor cells frequently express a lipogenic phenotype and how this may influence the results observed in THP-1 monocytes should be addressed at some point.

Response:

We thank the reviewer for this comment to improve our manuscript. We have added a discussion of the differences from the previous study by Mattos et al in the Discussion section: (pages 20-21, lines 446-453).

“Further studies are needed to confirm these points. Other studies have reported that M. leprae infection promotes cholesterol accumulation rather than TAG [8]. This discrepancy may be due to the different MOI and cell systems used. The number of bacteria used in our study was 10 times higher than that in previous reports. Additionally, the cells used in the previous report were primary cultures of human macrophages. Primary cells infected with M. leprae are known to be rich in cholesterol and cholesterol esters, but not THP-1 cells [27].”

3. Figure 2.

From the literature it seems that GPAT1 and 2 are located in the mitochondria, while GPAT3 and 4 are related to LDs. It would be interesting to show the relative basal level of expression of each isoform in THP-1 monocytes. Is GPAT3 the more abundant isoform in this cell line?

Response:

In Figs 2A and B, we show the changes in GPAT1-4 mRNA levels relative to the control levels (MOI: 0). The endogenous GPAT isoform mRNA were expressed at a similar level prior to normalization, which is shown in S2 Fig. We described it in the text as “Endogenous GPAT isoforms were expressed at a similar level in THP-1 cells (S2 Fig).” (page 12, line 265). 

4. Figure 3. 

4-1. It would be interesting to show that KO cells of the other GPAT isoforms, specially GPAT4, are not related to TAG biosynthesis in the context of M. leprae infection. This would strengthen the data from the previous figure that only showed modulation of GPAT3 expression.

Response:

We thank the reviewer for pointing out this issue. As the reviewer suggests, GPAT4, like GPAT3, is a microsome enzyme with similar functions. However, as shown in Figs 2A-D, the only isoform induced by M. leprae infection was GPAT3. Furthermore, we confirmed that the expression of GPAT4 in GPAT3 KO cells did not change following M. leprae infection (S3 Fig). From these results, we consider GPAT3 to be the main isoform that responds to M. leprae infection. Studies on the potential role of other GPATs in M. leprae infection could be a subject of future studies.

To explain this, we added the following sentence to the text (page 14, lines 309-312):

“Although both GPAT3 and GPAT4 are microsomal enzymes that share similar functions, the expression of GPAT4 in GPAT3 KO cells did not change, even after M. leprae infection (S3 Fig), confirming that GPAT3 is the main isoform that responds to M. leprae infection (Fig 2).”

4-2. 3C- The quality of the microscopy images is not good. I would also suggest increasing the size.

Response:

We performed fluorescent staining using the LipidTOX reagent to visualize cellular lipid droplet and replaced the Oil red O staining shown in Fig 3C. The results show a clear difference in lipid droplets formation between Wild-type and GPAT3 KO cells. In addition, we responded to a request from reviewer 1 by evaluating the brightness of each cell using the ROI (region of interest) analysis with Olympus FV10i software (Fig 3D).

The Results section was amended as follows (page 14, lines 304-309):

“We then compared the effect of M. leprae infection on lipid droplet formation in wild-type and GPAT3 KO cells using LipidTOX staining (Fig 3C) and quantified the LipidTOX fluorescence intensity in cells (Fig 3D). We observed a large number of lipid droplets in wild-type cells following M. leprae infection, whereas the number of lipid droplets was reduced by about 80% in GPAT3 KO cells, compared to the wild-type cells.” The Materials and Methods and figure legend were corrected accordingly (page 8, lines 175-177 and page 15, lines 338-342).

4-3. 3E- The conclusion drawn from this experiment that M. leprae utilizes host TAG is not convincing. During cell disruption and M. leprae isolation, TAG molecules could nonspecifically bind to the hydrophobic bacterial cell envelope. This would be more likely to occur in WT cells where the levels of labeled TAG are higher. To rule out this possibility, the inclusion of a control in which heat-killed M. leprae is incubated with cell lysates prepared from monocytes pre-labeled for 24 h with stearic C14 is strongly recommended.

Response:

Based on the reviewer’s suggestion, we performed an experiment to evaluate the possibility of non-specific TAG binding to the surface of M. leprae purified from THP-1 cells. To do this, THP-1 cells were treated with PMA (20 ng/mL) for 24 h to promote lipid droplet formation, then incubated with 0.2 �Ci of [14C] stearic acid for 16 h. The cell lysate was sonicated, mixed with M. leprae, and incubated for 24 h. M. leprae was isolated and the extracted lipids were separated by TLC to evaluate radioactivity.

As shown in revised Fig 3G, TAG was not detected in the M. leprae fraction, suggesting that nonspecific binding of TAG to the surface of M. leprae is not significant. We described this result as follows (page 15, lines 321-327): 

“Although vigorous washing steps were repeated to isolate M. leprae, we attempted to rule out the nonspecific binding of TAG on the surface of M. leprae as a possibility. Wild-type cells were stimulated with PMA and treated with [14C] stearic acid. Cells were then collected, sonicated, and the lysate containing TAG was mixed with M. leprae for 24 h. However, a 14C signal was not detected in the TAG fraction prepared from M. leprae (Fig 3G), suggesting that the nonspecific binding of TAG on M. leprae is negligible.”

The figure legend was modified accordingly.

4-4. Also, based on the results shown in Fig 4, it would be expected a smaller number of bacilli being recovered from KO cells. So, why the signal of the hsp70 gene is similar in WT and KO cells?

Response:

In the experiment shown in the original Fig 3D (revised Fig 3E), total DNA was extracted from WT cells and GPAT3 KO cells infected with M. leprae prior to conventional PCR. Although conventional PCR is not quantitative, we repeated the PCR with lower cycle numbers and replaced the Fig to better illustrate the difference in the number of bacilli in each cell. We thank the reviewer for the comment.

4-5. Finally, a second TLC showing the levels of labeled TAG in the host cell fraction will nicely complement the data. There are two slight bands above TAG in the WT infected cells that are also not seen in the GPAT3 KO. Would the authors believe that these are products derived of TAG utilization by M. leprae or other possible products related to GPAT3?

Response:

We agree with the reviewer that it would be nice to show labeled TAG in the THP-1 cell fraction after M. leprae infection. However, as stated above (response to 4-3) vigorous washing steps using detergent must be repeated to isolate M. leprae in the cells. Therefore, it is difficult to obtain a pure cellular fraction that does not contain M. leprae. Newly synthesized TAG was shown in Fig 3G following PMA treatment instead of M. leprae.

As the reviewer suggested, two faint bands are above the TAG bands, although the nature of these bands is not clear. Since the same bands were not detected in the GPAT3 KO fraction, it is likely that the signal was derived from lipid metabolites in the GPAT3 pathway, although confirmation is needed. We described this in the Discussion as follows (page 21, lines 453-456):

“Two additional bands can be seen above TAG spot in wild-type cells (Fig 3F) that could be metabolites of the TAG synthesis pathway, but the nature of these moieties is not yet clear.” 

5. Figure 4.

5-1. Figs A-C show that GPAT3 KO is apparently affecting M. leprae internalization, since less bacteria is seen inside the cells at 24 h of infection. The reduced number of bacteria in KO cells could also be related to a decrease in mycobacterial viability. So, the inclusion of earlier time points of infection in the analysis shown in Figs A-C, such as 4-6 h, is imperative and will definitively discriminate between these possibilities. If KO cells show less phagocytic capacity, as described in an earlier study referred in the discussion, this will absolutely compromise the conclusions drawn from Figs 3 and 4. The reduction in LDs accumulation in the KO cells could be because less bacteria is infecting the cell and therefore less modulation of lipid metabolism is seen and not so much because of the absence of GPAT3.

Response:

According to the reviewer’s suggestion, we performed a flowcytometric analysis between 3 and 24 h. The results show that a similar engulfment of M. leprae occurs in both WT and KO cells by 6 h, however, the number of cells decreases between 9 and 24 h as shown in revised Fig 4B. These results suggest that GPAT3 is not essential for the phagocytic activity of THP-1 cells, but rather important for M. leprae to stay within the cells. We described this in the Results as follows (page 16, lines 361-366):

“More than 80% of THP-1 cells were fluorescent at 3 h post-infection and the levels of such cells were similar between wild-type and GPAT3 KO cells until 6 h (Fig 4C). The FITC-positive cells began to decline at 9 h in GPAT3 KO cells and were reduced to 15.5% by 24 h. Conversely, the fluorescence of wild-type cells remained high (>80%) at 24 h (Fig 4C).”

We thank the reviewer for this suggestion that strengthened our manuscript. 

5-2. Concerning Fig D, a detailed explanation of how bacterial viability is determined is missing in the methods section. In this case, different from Fig 3, they mention that mRNA instead of DNA of the hsp70 gene was measured. Why?

Response:

We thank the reviewer for pointing out this issue. We have used pseudogene-derived RNA as an indicator of viability based on our previous observations. hsp70 was used as a control since its expression is rather stable. We explained this in the Result section and provided additional methods in the Materials and method section. 

“We previously reported that pseudogenes and noncoding regions in live M. leprae are not silent but instead are strongly transcribed and can be detected as RNA [14, 18, 24, 25]. The expression levels of these RNA are variable among patients with lepromatous leprosy [18], and decrease after antibiotic treatment (unpublished data). Therefore, we used the expression levels of these RNA to estimate the viability of M. leprae.” (page 17, lines 369-374)

“Quantification of M. leprae RNA expression

THP-1 cells (3 × 106) were infected with live M. leprae (MOI: 50) for 24 h and M. leprae was purified as described above. RNA was extracted using an RNeasy Plus Mini Kit (Qiagen) and reverse-transcribed to cDNA using a High Capacity cDNA Reverse Transcription Kit (Applied Biosystems) as described previously [18]. PCR primers to amplify cDNAs are listed in S1 Table. Touchdown PCR was performed using a Thermal Cycler Dice (Takara) as described previously [7, 19]. The PCR products were analyzed by 2% agarose gel electrophoresis.” (pages 9-10, lines 205-212)

5-3. In these experiments they are also using a MOI of 200. Why?

Response:

We thank the reviewer for this comment because we should have explained this in the text. The overall effects of M. leprae observed in in vitro conditions are dose-dependent up to a MOI 200 as shown in Fig 4D, but we usually use an MOI <20 that is closer to in vivo situations. However, to more clearly visualize intracellular M. leprae and obtain sufficient RNA for RT-PCR analysis, the MOI 200 was chosen for these experiments. We briefly explained this in the text as follows (page 16, lines 356-357):

“The MOI 200 was used to better visualize the results.”

5-4. I couldn’t follow whether hsp70 was used to normalize potential differences in bacterial loads between WT and KO cells. I would advise to do a qRT-PCR, which is indeed more quantitative to evaluate viability percentages. Actually, in several other studies M. leprae viability has been determined by the 16S rRNA/16S rDNA ratio as described by Martinez AN et al, J Clin Microbiol. 2009; 47(7):2124–30.

Response:

We agree with the reviewer that qRT-PCR would better provide quantitative data. However, as stated in the response to 5-2, we used hsp70 as a control, since its expression is rather stable in M. leprae. We have not used hsp70 mRNA levels to normalize the levels of others. We believed the results of Fig 4D show a clear difference between stable hsp70 expression and vulnerable expression of pseudogene-derived RNA. To clarify these points, we have modified the sentence in the Results section as follows (page 17, lines 377-379):

“The expression of M. leprae hsp70 mRNA was used as a control, since its expression level is stable [25] and did not change in both cells. These results suggest that the viability of M. leprae was lower in GPAT3 KO cells.”

6. Discussion

This section will need an extensive revision after the inclusion of new experiments as pointed out before. So far, the conclusions taken based on the assays with the GPAT3 KO cells are not correct. The authors should also take into account the limitations imposed by their in vitro model based on THP-1 cells and discuss the discrepancies observed between their results and those generated by others along recent years. Previous studies have shown that in context of human primary monocytes, both live and dead M. leprae induces LDs accumulation. Also, infected primary monocytes were found enriched in cholesterol and cholesterol ester LDs, finding not confirmed in THP-1 cells.

Response:

According to the reviewer’s comment, we have extensively revised the Discussion section by discussing the new experiments, mentioning the limitations of in vitro conditions using THP-1 cells, and citing previous reports using different cell systems. We believe that these modifications significantly improved our manuscript. We thank the reviewer for these comments.

---

## [Decision Letter · Decision Letter 1]

17 Feb 2021

PONE-D-20-21795R1

Mycobacterium leprae promotes triacylglycerol de novo synthesis through induction of GPAT3 expression in human premonocytic THP-1 cells

PLOS ONE

Dear Dr. Suzuki,

Thank you for submitting your manuscript to PLOS ONE. After careful consideration, we feel that it has merit but does not fully meet PLOS ONE’s publication criteria as it currently stands. Therefore, we invite you to submit a revised version of the manuscript that addresses the points raised during the review process.

We look forward to receiving your revised manuscript.

Kind regards,

Delphi Chatterjee

Academic Editor

PLOS ONE

Reviewers' comments:

Reviewer's Responses to Questions

**Comments to the Author**

1. If the authors have adequately addressed your comments raised in a previous round of review and you feel that this manuscript is now acceptable for publication, you may indicate that here to bypass the “Comments to the Author” section, enter your conflict of interest statement in the “Confidential to Editor” section, and submit your "Accept" recommendation.

Reviewer #1: All comments have been addressed

Reviewer #2: (No Response)

2. Is the manuscript technically sound, and do the data support the conclusions?

Reviewer #1: Yes

Reviewer #2: Partly

3. Has the statistical analysis been performed appropriately and rigorously? 

Reviewer #1: Yes

Reviewer #2: Yes

4. Have the authors made all data underlying the findings in their manuscript fully available?

Reviewer #1: Yes

Reviewer #2: Yes

5. Is the manuscript presented in an intelligible fashion and written in standard English?

Reviewer #1: Yes

Reviewer #2: Yes

6. Review Comments to the Author

Reviewer #1: (No Response)

Reviewer #2: The authors have responded to most comments and made the appropriate changes in the manuscript. However, few additional changes need to be included in the final version, as follows:

1-Authors should include details of the assay performed with dead bacteria to test the potential unspecific adsorption of TAG to M. leprae in the Materials and Methods section, such as the number of cells and bacteria used.

2-In figure 3C, imagens of uninfected wild type and KO cells stained with lipidTOX should be included.

In the Discussion seccion:

3-The sentence “In this study, we performed a comprehensive lipid analysis using HPTLC to show that TAG is the main lipid that accumulates in lipid droplets in M. leprae-infected macrophages” needs to be changed. Their study did not perform “a comprehensive lipid analysis “. Moreover, as pointed out in the first round of review, macrophages should be replaced by THP-1 cells.

4-In the sentence “Primary cells infected with M. leprae are known to be rich in cholesterol and cholesterol esters, but not THP-1 cells [27]”, reference 27 refers just to the first part, not to THP-1 cells. Please make changes accordingly.

5- In the section where they speculate about the possible involvement of PPAR�, they should mention the study by Diaz Acosta CC et al. 2018, in which they show that this transcriptional factor is induced by M. leprae in infected Schwann cells and participates in host cell lipid accumulation.

7. PLOS authors have the option to publish the peer review history of their article (what does this mean?). If published, this will include your full peer review and any attached files.

Reviewer #1: No

Reviewer #2: No

---

## [Author Response · Author response to Decision Letter 1]

9 Mar 2021

Reviewer 2

The authors have responded to most comments and made the appropriate changes in the manuscript. However, few additional changes need to be included in the final version, as follows:

Response: 

We thank the reviewer for the comments. We have addressed the reviewer’s concerns at below.

1. Authors should include details of the assay performed with dead bacteria to test the potential unspecific adsorption of TAG to M. leprae in the Materials and Methods section, such as the number of cells and bacteria used.

Response: 

We have included the experiment performed with wild-type THP-1 cells infected with dead M. lepra in S4 Fig, as well as the experiment using live M. leprae incubated with PMA-treated THP-1 cell lysate in Fig. 3G. Detailed protocol was described in the Material and Methods and in Figure legends accordingly as follows.

Materials and Methods, page 9, lines 189-191: 

“THP-1 cells (1 × 106) were inoculated with ether live or heat-killed M. leprae (MOI: 10 and 50) for 24 h, then incubated with 0.2 uCi [14C] stearic acid (American Radiolabeled Chemicals, Saint Louis, MO) for 16 h at 37°C with 5% CO2.”

Fig. 3 legend, pages 15-16, lines 343-345:

“(E and F) WT and GPAT3 KO THP-1 cells were cultured in medium containing 0.2 uCi of [14C] stearic acid for 16 h after M. leprae infection (MOI: 10 and 50).”

S4 Fig legend, page 30, lines 664-666:

“Wild-type THP-1 cells were inoculated with either live or heat-killed M. leprae (MOI: 10 and 50), then cultured with 0.2 uCi of [14C] stearic acid for 16 h. M. leprae was isolated and extracted lipids were separated by TLC to evaluate radioactivity.”

2. In figure 3C, images of uninfected wild type and KO cells stained with lipidTOX should be included.

Response: 

According to the suggestion, we added images of the LipidTOX fluorescent staining of uninfected wild type and KO cells in Fig. 3C. The Figure legend was modified accordingly.

3. The sentence “In this study, we performed a comprehensive lipid analysis using HPTLC to show that TAG is the main lipid that accumulates in lipid droplets in M. leprae-infected macrophages” needs to be changed. Their study did not perform “a comprehensive lipid analysis “. Moreover, as pointed out in the first round of review, macrophages should be replaced by THP-1 cells.

Response: 

According to the Reviewer’s suggestion, we have rephrased a sentence in the Discussion section as follows (page 19, lines 407-409):

“In this study, we performed a qualitative lipid analysis using HPTLC to show that TAG is the main lipid that accumulates in lipid droplets in M. leprae-infected THP-1 cells.”

4. In the sentence “Primary cells infected with M. leprae are known to be rich in cholesterol and cholesterol esters, but not THP-1 cells [27]”, reference 27 refers just to the first part, not to THP-1 cells. Please make changes accordingly.

Response: 

We have corrected the position of the reference as follows (pages 20-21, lines 452-456).

“Additionally, the cells used in the previous report were primary cultures of human macrophages, and primary cells infected with M. leprae are known to be rich in cholesterol and cholesterol esters [27]. Differences in the lipid moieties accumulated in response to M. leprae infection between primary cells and THP-1 cells need to be clarified in the future.”

5. In the section where they speculate about the possible involvement of PPARg, they should mention the study by Diaz Acosta CC et al. 2018, in which they show that this transcriptional fact or is induced by M. leprae in infected Schwann cells and participates in host cell lipid accumulation.

Response:

We thank the reviewer for pointing out this issue. We have cited the suggested paper (Diaz Acosta CC et al., 2018) as well as ours (Luo, et al., 2020) describing the involvement of PPARg in M. leprae infection in the Discussion section as follows (page 21, lines 466-471):

“Recently, we have reported that activation of PPARg and PPARd is important for lipid accumulation in M. leprae-infected THP-1 cells [Luo, 2020 #62]. In Schwann cells, phenolic glicolipid-1 (PGL-1) of M. leprae promoted lipid droplet formation by activating crosstalk between CD206 and PPARg [Diaz Acosta, 2018 #32]. Therefore, M. leprae might utilize the signal transduction pathway(s) mediated by PPARg to induce GPAT3 expression in infected cells.”

---

## [Editor Report · Decision Letter 2]

15 Mar 2021

Mycobacterium leprae promotes triacylglycerol de novo synthesis through induction of GPAT3 expression in human premonocytic THP-1 cells

PONE-D-20-21795R2

Dear Dr. Suzuki,

We’re pleased to inform you that your manuscript has been judged scientifically suitable for publication and will be formally accepted for publication once it meets all outstanding technical requirements.

Kind regards,

Delphi Chatterjee

Academic Editor

PLOS ONE
---

## [Editor Report · Acceptance letter]

18 Mar 2021

PONE-D-20-21795R2 

*Mycobacterium leprae* promotes triacylglycerol *de novo* synthesis through induction of GPAT3 expression in human premonocytic THP-1 cells 

Dear Dr. Suzuki:

I'm pleased to inform you that your manuscript has been deemed suitable for publication in PLOS ONE. Congratulations! Your manuscript is now with our production department. 

Kind regards, 

on behalf of

Dr. Delphi Chatterjee 

Academic Editor

PLOS ONE